# Femtosecond Laser Micro-/nano-texturing of Stainless Steels for Surface Property Control

**DOI:** 10.3390/mi10080512

**Published:** 2019-07-31

**Authors:** Tatsuhiko Aizawa, Tadahiko Inohara, Kenji Wasa

**Affiliations:** 1Surface Engineering Design Laboratory, Shibaura Institute of Technology, Tokyo 144-0045, Japan; 2LPS-Works. Co., Ltd, Tokyo 144-0033, Japan; 3MicroTeX Labs, llc., Tokyo 144-0051, Japan

**Keywords:** micro-/nano-texturing, femtosecond laser machining, stainless steel substrates, surface geometry, fractal dimension, aspect ratio, contact angle

## Abstract

Surface geometry has had an influence on the surface property, in addition to the intrinsic surface energy, of materials. Many physical surface modification methods had been proposed to control the solid surface geometry for modification of surface properties. Recently, short-pulse lasers were utilized to perform nano-texturing onto metallic and polymer substrates for the improvement of surface properties. Most of the papers reported that the hydrophilic metallic surface was modified to have a higher contact angle than 120–150°. Little studies explained the relationship between surface geometry and surface properties. In the present study, the laser micro-/nano-texturing was developed to describe this surface-geometric effect on the static contact angles for pure water. Micropatterns with multi spatial frequencies are designed and synthesized into a microtexture. This tailored microtexture was utilized to prepare for computer aided machining (CAM) data to control the femtosecond laser beams. The nano-length ripples by laser induced periodic surface structuring (LIPSS) supposed onto this microtexture to form the micro-/nano-texture on the AISI304 substrate surface. Computational geometry was employed to describe this geometric profile. The fractal dimension became nearly constant by 2.26 and insensitive to increase of static contact angle (θ) for θ > 150°. Under this defined self-similarity, the micro-/nano-textured surface state was controlled to be super-hydrophobic by increasing the ratio of the highest spatial frequency in microtextures to the lowest one. This controllability of surface property on the stainless steels was supported by tailoring the wavelength and pitch of microtextures. Exposure testing was also used to evaluate the engineering durability of this micro-/nano-textured surface. Little change of the measured fractal dimension during the testing proved that this physically modified AISI304 surface had sufficient stability for its long-term usage in air.

## 1. Introduction

The surfaces of materials have unique characteristics, different from their body in nature [1]; e.g., a raindrop does not swell on the lotus leaf at the presentation of fine, thin trichomes, and, a water strider skates on the water surface by way of textured feet. The surface property of solid materials is characterized by the surface energy (E) [2]; e.g., when E is high, the liquid drop easily covers the substrate surface. On the other hand, the drop is isolated on the substrate when E is low. Instead of the direct measurement of E, the swelling angle or the contact angle (θ) is often employed as an indicator of surface energy [3]. That is, cos (θ) approaches 1.0 when E is the highest; E becomes the lowest when cos (θ) goes to −1. Following this, the surface state is hydrophilic when cos (θ) > 0 while it becomes hydrophobic when cos (θ) < 0. In particular, the surface with cos (θ) ~ 1 is called to be super-hydrophilic, while that with cos (θ) ~ −1, super-hydrophobic [4]. 

The surface chemical state as well as the surface physical state have much influence on this contact angle or the surface energy [5]. As demonstrated in [6,7], the contact angle is tunable from the super-hydrophilic state to the super-hydrophobic one by the molecular surface modification as well as the surfactants. Since the first finding on laser-induced periodic surface structuring (LIPSS) by laser machining, many studies on the surface physical modification have been reported in the literature [8,9,10,11,12]. In particular, the femtosecond laser nano-texturing is utilized to simply modify the material surface properties from hydrophobic state to super-hydrophobic one. There are still many engineering issues left; e.g., the absence of a theoretical estimate on the controllability of surface conditions, and durability of physically modified surfaces by way of laser treatment. The authors have proposed surface profile control by way of synthesizing the microtextures by direct laser machining and LIPSS-ripples [13,14]. This method makes full use of spatial multi-frequencies to modify the original two dimensional surface to the textured one with self-similarity and a local aspect ratio [14]. With aid of the low temperature plasma nitriding without precipitation of nitrides, this method was applied to physical modification of a die surface for hot stamping of oxide glasses and optical polymers [15,16]. 

In the present paper, the austenitic stainless steel AISI304 substrate surface is physically modified to be hydrophobic or super-hydrophobic by micro-/nano-texturing with use of the femtosecond laser machining system. The computational geometry is utilized to synthesize the multi-frequency micro-textures, and, to evaluate the micro-/nano-textures, laser-machined onto the AISI304 surface. Its surface is actually modified to have tailored micro-textures with superposition of nano-structured ripple on them. This controllability of micro-/nano-textures is preferable to their industrial applications. The computational geometry proves that they have self-similarity with the fractal dimension in the order of 2.2 to 2.3 and that the ratio of higher spatial frequency to lower one in the microtextures has influence on increase of the contact angle. Optimum design of this micro-nanotextures provides a way of effective physical surface modification toward super-hydrophobicity with higher contact angles. In addition, the durability of these controlled super-hydrophobic surfaces is demonstrated by the long-term exposure test in air.

## 2. Experimental Procedure

The femtosecond laser machining system was employed to make micro-/nano-texturing to stainless steel substrates for physical modification of surface properties. The computational geometry was utilized to make texture design with multiple spatial frequencies for femtosecond laser beam control. Besides for the static and dynamic contact angle measurement, the surface profile of textures substrates was measured to quantitatively describe the surface geometry.

### 2.1. Femtosecond Laser Machining

A femtosecond laser machining system is employed for physical modification of metallic and ceramic surfaces with use of LIPSS. Figure 1a depicts the femtosecond laser machining system in the present study.

Since the focused spot of work materials is subjected to ultra-high-power irradiation, how to scan the beam spot becomes more important when using this laser machining system. Higher repetition frequency as well as higher scanning speed are needed for fast-rate, dimensionally accurate machining. In general, there are two methods to form the tailored textures with multiple spatial frequencies onto the work material surface by the laser irradiation; e.g., the direct laser machining and the time-sequence control of laser pulses. In this study, the former approach was employed for simplicity. Since the laser spot size is 1 μm in the present system, the spatially synthesized profile is directly digitized into computer aided machining (CAM) data to control the laser-machining paths. In this case, higher frequency components are formed by LIPSS through the interaction between the incident femtosecond laser beam and its scattering one by the micro-textured surface profile. As experimentally demonstrated in later, these LIPSS ripples are superposed onto the micro-textured profiles.

There are also two approaches to control the micro-textures in the direct laser machining; e.g., uniaxial machining and biaxial machining. In the former, the high frequency texture components superpose onto the uniaxial fundamental texture. In the latter, these high frequency modes are mixed with biaxial fundamental texture. In the following experiments, both methods are utilized to form the synthesized surface textures.

### 2.2. Micro-/Nano-Texturing Design

Micro-texturing design stands on how to synthesize the spatial micropatterns with different pitch (Λ) and shift (φ). A targeting microtexture is shown in Figure 2. This texture is constructed by synthesizing three micropatterns; e.g., the first texture with Λ = 20 μm and φ = 0 μm, the second one with Λ = 20 μm and φ = 10 μm and the third one with Λ = 40 μm and φ = 20 μm. In the practical laser micro-texturing, the beam intensity is controlled to attain the targeting height (H_max_) of synthesized microtexture; e.g., H_max_ = 12 μm in Figure 2. As stated in 2.1., this designed microtexture is laser machined onto the surface as the uniaxial and biaxial fundamental profiles, respectively.

In the nano-texturing by LIPSS, its spatial pitch (Λ_n_) is predetermined by the wavelength of the femtosecond laser to be used; e.g., in the present case, Λ_n_ = 100 to 200 nm. To be discussed later, this nanotexture can be superposed onto the synthesized microtexture on the laser-processed surface. Owing to this superposition, the laser processed surface has unique self-similarity with the well-defined fractal dimension.

### 2.3. Computational-Geometry Analysis

In mechanical engineering, the surface profile is evaluated by several roughness values; e.g., the average roughness, Ra, for classification of surface roughness levels, and, the maximum roughness, Rz, for mechanical evaluation on the effect of roughness on the friction and wear. The surface property is controlled not only by this ensemble-averaged geometric change but also by the self-similarity as well as the local aspect ratio [14].

A flat surface is completely two dimensional at the absence of microstructures. Assuming that a finer structure is formed repetitively by similar processing, the surface is expected to have self-similarity; e.g., the current self-similar unit consists of b self-similar sub-units when contacting it by 1/a. The fractal dimension (D), which is defined by log (b)/log(a), becomes a suitable parameter to describe this complex surface after repetitive contraction. Using the measured D, the micro-/nano-textured surface is characterized by 2 < D < 3. There are no definite unit sizes of a and b in practical surface geometries; hence, the fractal dimension must be estimated from the measured surface profiles. The box counting method [17] is a powerful tool to automatically determine D from optical microscopy and SEM images of surface profile.

Consider the intensity profile g (x, y) on each unit of pixels in the SEM image of surface with the digitizing unit length h. Reducing this h, the approximate fractal dimension D_h_ is calculated for each h by: D_h_ = −log(g(h))/log(h)(1)
where g(h) is integrated intensity of measured profile on the cell area with h × h. D is estimated by the limit of D_h_ by h → 0. In practical calculation, the measured g(h) is logarithmically plotted against h; the estimated gradient of this plot results in the accurate estimate of D.

The surface property is also sensitive to the local aspect ratio in surface textures. In the present laser treatment, the maximum local aspect ratio is directly controlled by the tailored micro-texture profile. The longitudinal aspect ratio is defined by the ratio of height to pitch in the fundamental textures in the laser irradiation.

This synthesizing method allows us to make another controlling approach of the surface geometry. The characteristic frequency, f_H_, of the induced high frequency components to synthesized texture, is independently controlled from the basic frequency, f_0_, in the fundamental texture. The ratio of f_H_ to f_0_ plays a role to be working as other geometric parameters rather than the fractal dimension.

### 2.4. Observation, Measurement and Work Materials

The austenitic stainless steel type AISI304 sheets and plates were employed as a work material for the laser micro-/nano-texturing. SEM (Scanning Electron Microscopy; Richo, Co., Ltd., Tokyo, Japan) was utilized to measure the surface textures. The static contact angle was measured by using the MCA-3 system (Kyowa Interface Science, Co., Ltd., Saitama, Japan) and the projector (Nicon, Co., Ltd., Tokyo, Japan), respectively.

## 3. Experimental Results

The AISI304 substrate surface was micro-/nano-textured to make its surface characterization and to physically modify the surface property. This textured surface was further controlled to have super-hydrophobicity with sufficient durability in atmosphere.

### 3.1. Micro-/Nano-Textured AISI304 Substrate

The deigned microtexture in Figure 2 was formed onto the AISI304 substrate surface by the femtosecond laser machining. Figure 3 depicts the scanning electron microscopy (SEM) image on the uniaxially micro- /nano-textured surface. Nano-textures by LIPSS superposed onto the microtextures, which were directly machined by the femtosecond laser irradiation.

The cross-sectional textured surface profile was data-processed from this two-dimensional SEM image to provide for further data processing and to estimate the fractal dimension (D) of this micro-/nano-texture formed by femtosecond laser machining. Figure 4 shows the one-dimensional surface profile selected in the lateral direction from the two-dimensional image in Figure 3. A single spatial texture a→b→c→d→e in Figure 4 just corresponds to A→B→C→D→E in Figure 3.

In correspondence to the superposition of nanotextures onto the micro-textures in Figure 3, high frequency peaks are detected on the main profile of microtextures. The maximum height reached to 13 μm, a little higher than H_max_ = 12 μm in design. The spatial pitches in Figure 4 is equal to the designed data in Figure 2. That is, the tailored micro-/nano-pattern in computer-aided design (CAD) is reproduced as an actual micro-/nano-textured surface on the AISI304.

### 3.2. Relationship Between the Aspect Ratio and the Contact Angle

This micro-/nano-textured surface is expected to have surface properties modified from the flat AISI304 surface. Although most of studies [8,9,10,11,12] stressed the importance on the nanostructure by LIPSS to affect the surface property, the geometric effect of synthesized microtextures in Figure 2 and Figure 4 on the contact angle was first investigated to describe the physical modification by the micro-/nano-texturing. Figure 5 depicts the relationship between the aspect ratio in the micro-texture and the static contact angle. This aspect ratio of the lowest peak height to the highest one in the microtexture in Figure 4 is 0.24. The static contact angle is measured to be 170°. That is, the geometric design on the microtexture is also important for effective surface modification.

In the range of aspect ratio from 0.10 to 0.30, almost all the measured contact angles become higher than 150°; the surfaces in this range are super-hydrophobic. This proves that the micro-/nano-texturing changes the original hydrophilic surface to a super-hydrophobic one with a higher contact angel than 150° irrespective of the aspect ratio, but, that this aspect ratio becomes a key to improve this super-hydrophobicity with the contact angle toward 170° as well as the repellency of pure water.

### 3.3. Computational-Geometry Analysis on the Textured Surface

Two geometric parameters in the textured surface are analyzed to investigate their relationship with the measured contact angle. As before mentioned, the fractal dimension (D) is estimated by the box-counting method also from the SEM-image in the following. The surface height integral, g(h), is calculated for each unit cell with the width of h in Figure 4. Figure 6 depicts the relationship between log(g_h_) and h. 

From this regression line in Figure 6, the fractal dimension on the cross-section of the texture is determined to be 1.26 from the gradient. The fractal dimension of original micro-/nano-textured surface in Figure 3 is given by D = 1. 26 + 1 = 2.26. 

After one- and two-dimensional Fourier transformations, both the highest spatial frequency (f_H_) and the fundamental frequency (f_0_) were calculated. This ratio of f_H_ to f_0_ also distributes in the micro-/nano-textured surface so that f_H_/f_0_ ratio has another influence on the measured contact angle.

### 3.4. Controllability of Surface Prperties by Geometric Parameters

The screened data in Figure 5 on the relationship between the aspect ratio and the contact angle suggests the important effect of geometric parameters on the surface property. This sensitivity of the contact angle to measured D and f_H_/f_0_ ratio for each micro-/nano-texture is investigated to search for a way of optimization in surface geometry toward the super-hydrophobicity.

First, the variation of measured contact angle with the fractal dimension is studied in Figure 7. In the transient state from hydrophobic to super-hydrophobic states, the fractal dimension becomes nearly constant by D = 2.26. After this [18], most of the hydrophobic surfaces with a higher contact angle than 100° have the fractal dimension of 2.16 < D< 2.29. This implies that every micro-/nano-texture with a higher contact angle than 160° has the same self-similarity as defined in each material system and that D has little impact to increase the contact angle.

Next, the ratio of f_H_/f_0_ was selected to investigate the variation of maximum contact angle with this ratio. The noisy data were included by geometric irregularities in the micro-nano-textured surface with lower contact angles as suggested in [19]. This ratio of f_H_/f_0_ increases monotonously with the contact angle as shown in Figure 8. That is, higher frequency components in the surface geometry have more important role to improve the contact angle toward super-hydrophobicity even in the same self-similarity of surface texture. The transition from hydrophobicity to superhydrophobicity is described by nano-texture formation on the microtexture in the fractal surface.

### 3.5. Durabiility in Controlled Surfaces by Micro-Nano-Texturing

Mechanical elements and parts with physically modified surfaces by micro-/nano-texturing are expected to experience various surface contamination by surfactants. There were lots of papers on the effect of contamination on the change of contact angles. The cellulose based organic films were synthesized on the titanium surface by the femtosecond laser irradiation [20]; the surface property was affected by this insitu formed organic layer. The surface property changed from super-hydrophobicity to super-hydrophilicity at the absence or presence of organic surfactant film [21]. Those studies worried about the durability of hydrophobic or super-hydrophobic surfaces formed by the femtosecond laser texturing.

The exposure test was performed to investigate the geometric profile change of super-hydrophobic surface formed by the present method with the exposure time. The test-piece was exposed to air in atmospheric condition with the average humidity of 40%. An SEM image was taken at the specified time after air-blowing the surface for measurement of the fractal dimension. The contact angle was measured at the beginning and ending of test; e.g., θ = 170.2° at t = 0 and θ = 168.5° at t = 26,050 s. Figure 9 depicts the variation of the fractal dimension with exposure time. The measured fractal dimension is held to be nearly constant by 2.25 for 20,000 minutes or 330 hours. This proves that the super-hydrophobic AISI304 surface has sufficient durability after long-term usage with exposure to air in an atmospheric condition. 

In fact, the contact angle of test-piece surface after exposure test for 26,050 minutes was still 164°; the super-hydrophobicity was preserved after the exposure test.

## 4. Discussion

The durability of physically modified surfaces using the present method is enhanced by combining them with low temperature plasma nitriding [22]. The nitrogen stainless steels with a higher nitrogen solute content than 4 mass % have high hardness and corrosion toughness in their thick surface layer up to 0.1 mm. As reported in [16], the physically modified surface after this plasma nitriding worked as a hot stamping die to transcribe the hydrophobic surface geometry to oxide glasses with the glass transition temperature of 923 K.

High controllability of surface geometry by way of physical modification, demonstrates that the microstructure surface in design can be tuned into the actual micro-/nano-textured surface, as tailored. A hybridized artificial surface with hydrophilic and hydrophobic states half-in-half as well as the designed anisotropy in contact angle can be put into practice for each application of the functional surface. In particular, micro-devices as well as micro-electro-mechanical systems (MEMS) require for the functionalized surface; e.g., a micro-fluidic chip [23] needs the selectively hydrophilic on the wall surfaces of channels, and, a micro-cooling device [24] favors the textured surface for efficient boiling heat transfer. 

How to accommodate the hydrophilic or hydrophobic sub-surfaces onto the mechanical elements or parts becomes an issue of engineering; e.g., the selective surface modification onto the channel walls in the micro-fluidic chips and the heat transfer walls in micro-cooling device requires for the excellent control of short-pulse laser beams [13]. In particular, a nozzle with small outlets for precise ink-jet printing and bio-printing [25] must have hydrophobic or super-hydrophobic outlet surfaces to dispense fine-sized droplets onto the sheets at the controlled positions.

Nano-texturing process is controllable by the wavelength, the pulse duration and the repetition, even when using current femtosecond laser machining. After the recent advancements of semiconductor lasers [26,27], more flexible laser micro-/nano-texturing is available to control the surface properties of miniature nozzles as well as nozzle array.

## 5. Conclusions

The micro-/nano-texturing method is proposed as a femtosecond laser surface modification process for stainless steels. The microtextures with multi spatial frequencies with different wave lengths and pitches are designed for each application and synthesized into a controlling microtexture as CAM data for laser beam control in machining. Through LIPSS in the femtosecond laser texturing, the nano-scaled ripples induced by LIPSS superpose on the tailored microtextures to form the uniaxial or biaxial micro-/nano-textures on the AISI304 surface. The original hydrophilic state for a flat AISI304 surface changes to become hydrophobic and super-hydrophobic with less sensitiveness to the fractal dimension, but with increasing the ratio of highest spatial frequency to the lowest one in the microtextures. This controllability of surface properties by the micro-/nano-texturing stands on the well-defined correlation between the tailored micropatterns in CAD and the laser-machined microtextures on AISI304.

Engineering durability of surface geometry formed by this micro-/nano-texturing is also investigated by the exposure testing to air at the atmospheric condition. No essential difference of measured fractal dimension for 360 hours demonstrates that a micro-/nano-textured AISI304 surface has sufficient stability in applications. In particular, the present method with the aid of low temperature plasma nitriding provides a solution to transcribe the hydrophobic and super-hydrophobic surface state on the nitrided austenitic stainless steel die onto the oxide glasses as well as the metallic and polymer sheets through hot, warm and cold stamping.

## Figures and Tables

**Figure 1 micromachines-10-00512-f001:**
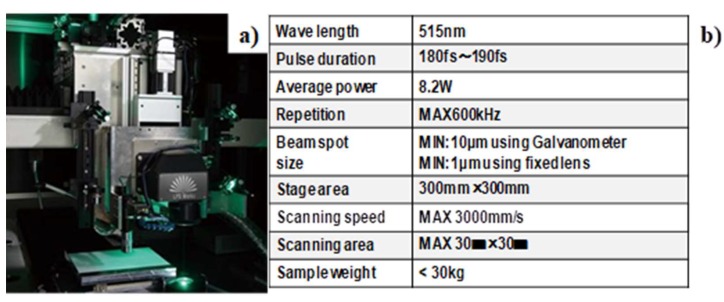
Femtosecond laser machining system for micro-/nano-texturing into AISI304 substrate surfaces. (**a**) Outlook of system, and (**b**) its fundamental features.

**Figure 2 micromachines-10-00512-f002:**
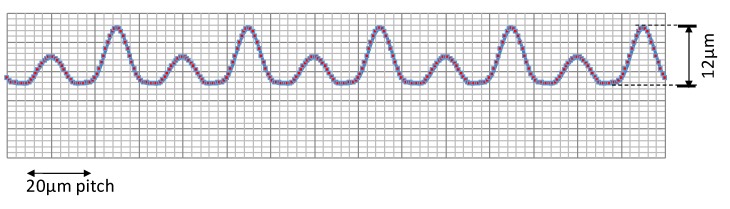
A synthesized microtexture for femtosecond laser microtexturing.

**Figure 3 micromachines-10-00512-f003:**
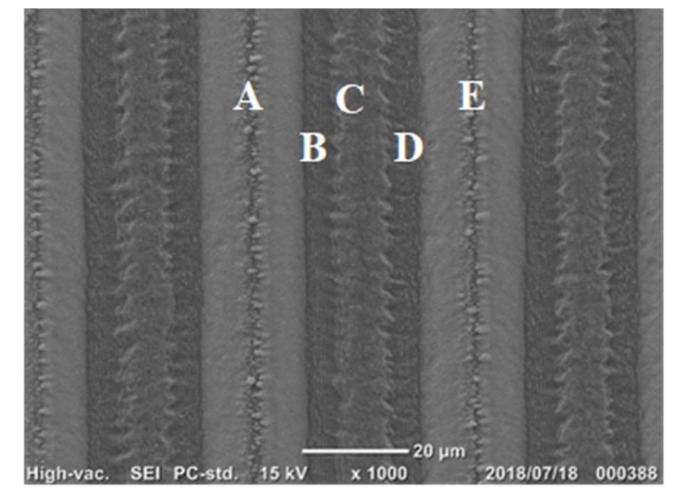
Uniaxially formed micro-/nano-textures onto the AISI304 surface by the femtosecond laser.

**Figure 4 micromachines-10-00512-f004:**
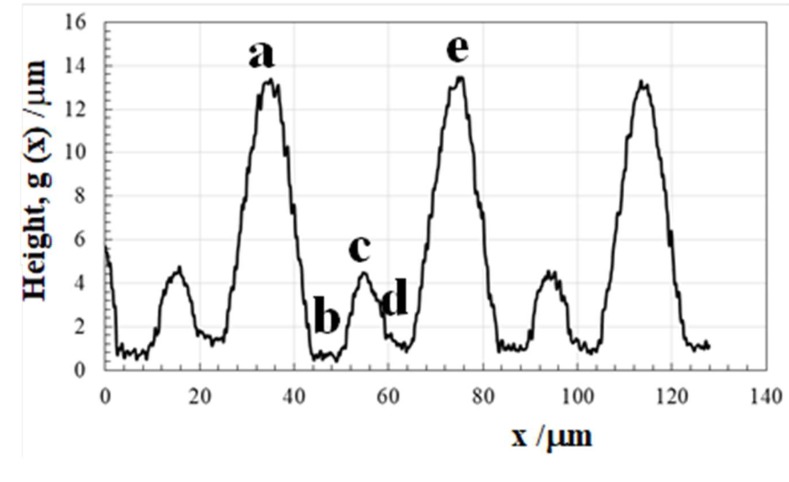
One-dimensional surface profile, selectively processed from the two-dimensional scanning electron microscope (SEM) image.

**Figure 5 micromachines-10-00512-f005:**
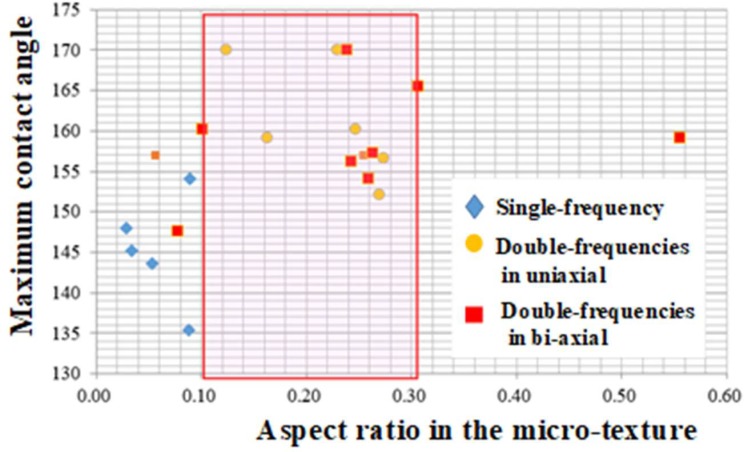
Effect of the aspect ratio on the measured static contact angle for various micro-textures. The red-meshed area indicates the high-lighted zone to yield super-hydrophobic surfaces in all.

**Figure 6 micromachines-10-00512-f006:**
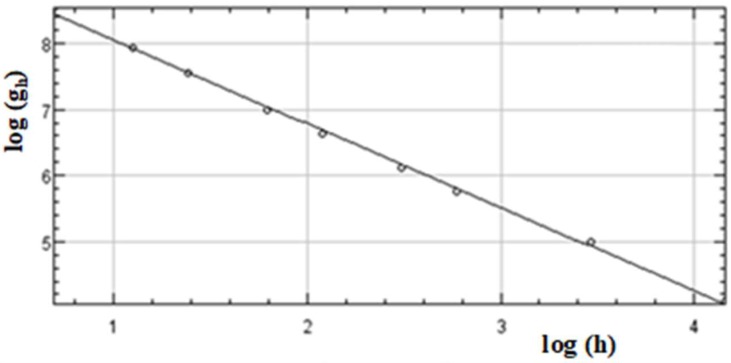
Relationship between the log(g_h_) and the unit cell length, h, for the micro-/nano-texture in Figure 4, after the box-counting method.

**Figure 7 micromachines-10-00512-f007:**
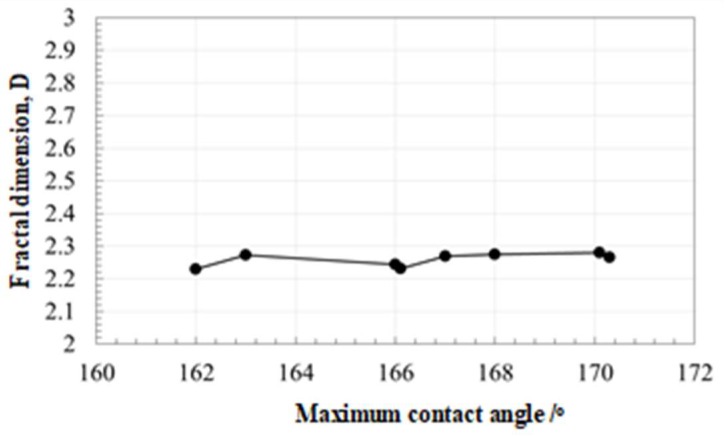
Relationship of the fractal dimension to the measured maximum contact angle.

**Figure 8 micromachines-10-00512-f008:**
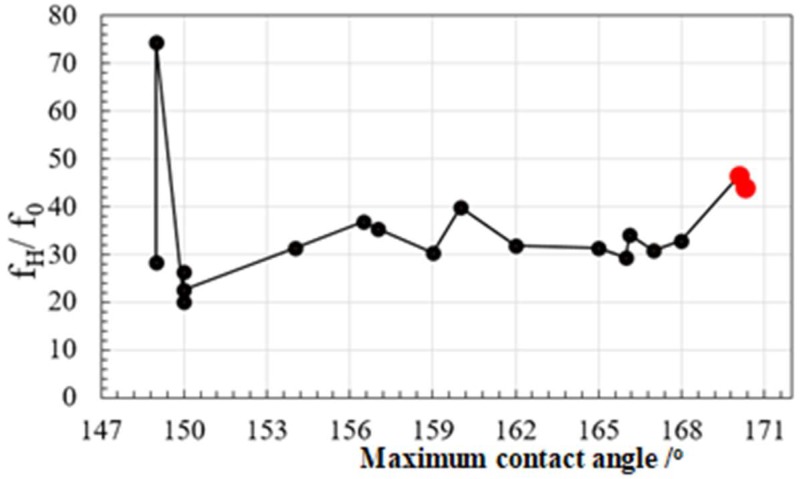
Relationship between the f_H_/f_0_ ratio and the maximum static contact angle. The red dots represent the highest contact angles by synthesizing the multi-frequency textures in Figure 2.

**Figure 9 micromachines-10-00512-f009:**
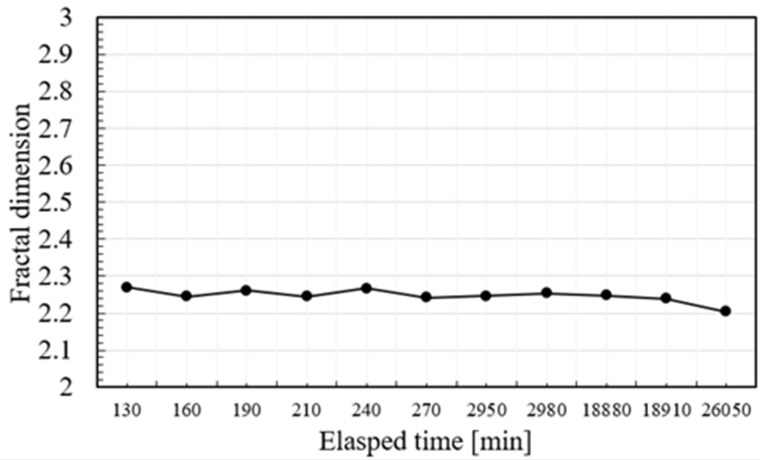
Variation of the fractal dimension measured from a SEM image of a test-piece exposed to air with exposure time.

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
