# Peer review of "Femtosecond Laser Micro-/nano-texturing of Stainless Steels for Surface Property Control"

_micromachines, 2019, doi:10.3390/mi10080512_

Round 1

Reviewer 1 Report

The submitted paper concern the properties of stainless steel surfaces modified by a femtosecond laser surface process. The super-hydrophobicity in relation to surface geometry and the durability of the treated surfaces are investigated.

The study is properly organized, but some revisions are necessary before the publication.

The following are my suggestions and remarks.

I suggest removing the keyword “hydrophobicity”, “super-hydrophobicity” is already adequate.

Please, rephrase the sentence at lines 85-84, the meaning is not clear.

Additional details about the experimental procedures should be given in section 2.4.

Line 202: Explain that an SEM image is shown in Figure 3.

In discussing the results reported in Figure 4, could you indicate the correspondence between the peaks and the dark/pale areas in fig. 3?

Figure 5: Add a legend to explain the different color and shape of the points in the graph; describe in the caption the meaning of the highlighted area.

Figure 7: Why only the results for contact angles from 162° to 171° are reported? It could be interesting to show the changes due to the transition “from hydrophobic to super-hydrophobic states”. The range reported in Figure 8 could be more adequate.

Figure 8: Explain meaning of the different color points.

Section 3.5: Explain the choice of the conditions to evaluate the durability of the treated surface. The exposure to the relative humidity of 40% for less than 20 days is not enough to assess the long-term performance and durability. In addition, report (in a table or in a graph) the contact angle values measuring in this test.

Section 4: This section seems a little off-topic. Most of the sentences seem more appropriate for the introduction, while the obtained results are not discussed in detail. Please, revise.

Author Response

My replies were summarized in the follwoing word file.

Reviewer 2 Report

Manuscript No. micromachines-550722  

Title: “Femtosecond laser micro-/nano-texturing of stainless steels for surface property control” 

This manuscript theoretically and experimentally designed a laser micro-/nano-texturing to describe the surface-geometric effect on the static contact angles for pure water. The author designed Micropatterns with multi spatial frequencies and synthesized into a microtexture. Their proposed tailored microtexture is utilized to prepare for CAM data to control the femtosecond laser beams. The nano-length ripples by LIPSS supposed onto this microtexture to form the micro-/nano-texture on the AISI304 substrate surface. The computational geometry is employed to describe their proposed geometric profile. The presented results are enough interesting for publication in micromachines journal. I recommend the paper to publish in this journal AFTER a minor revision in the references of the paper.

1. As you used Femtosecond laser in your design, you can also cite some new and updated papers about semiconductor lasers such as:
[Ref] “Development of optically pumped DBR-free semiconductor disk lasers” (Conference Presentation). In Vertical External Cavity Surface Emitting Lasers (VECSELs) VII (Vol. 10087, p. 100870K). International Society for Optics and Photonics. (2017).
[Ref] " Bandgap engineering and prospects for radiation-balanced vertical-external-cavity surface-emitting semiconductor lasers" Optics express 26 (10), 12985-13000 (2018).
[Ref] “16 W DBR-free membrane semiconductor disk laser with dual-SiC heatspreader.” Electronics Letters, 54(7), 430-432 (2018).
[Ref] "Frequency-Doubled VECSEL Employing a Volume Bragg Grating for Linewidth Narrowing." CLEO: QELS_Fundamental Science. Optical Society of America, (2018).
[Ref] " Time, space, and spectral multiplexing for radiation balanced operation of semiconductor lasers." Optics express 26 (18), 24124-24134 (2018).

[Ref] “Spin-orbit interaction effects on the electronic structure of spherical quantum dot with different confinement potentials,” Superlattices and Microstructures, Volume 111, November 2017, Pages 671-677 (2017).

Please add these references about new and updated laser research.

2. Please make Figure 6 more clear. Thank you.

3. The reference should be improved to some important references about the lasers in this work.

Overall, this is so interesting work. And I recommend this manuscript for publication in this journal after adding some new and updated references about the important part of the work.

Author Response

My rely was included in the following word file.
